# Tailored Psychological Interventions to Manage Body Image: An Opinion Study on Breast Cancer Survivors

**DOI:** 10.3390/ijerph20042991

**Published:** 2023-02-08

**Authors:** Valeria Sebri, Gabriella Pravettoni

**Affiliations:** 1Applied Research Division for Cognitive and Psychological Science, IEO European Institute of Oncology IRCCS, 20139 Milan, Italy; 2Department of Oncology and Hemato-Oncology, University of Milan, 20122 Milan, Italy

**Keywords:** breast cancer survivors, body image, injured self, psychological interventions

## Abstract

Objective: Oncological care affects the body strongly, even some years after therapies. Body image, as the mental representation of one’s own body, is particularly affected by breast cancer, with a high level of dissatisfaction and negative perception. Literature has shown the effectiveness of various psychological interventions to promote body image in breast cancer survivors, dealing with inner sensations and related emotions and thoughts. The present opinion study presents BI issues and personalized psychological interventions to increase a positive BI in breast cancer survivors. Conclusions: Implementing specific and personalized psychological interventions tailored on BI, the characteristics of oncological journey and emotional and cognitive issues is fundamental. Directions for clinical practice are given.

## 1. Introduction

Breast cancer diagnosis, interventions, and treatments (e.g., surgery, chemotherapy, radiotherapy, and hormonal therapy) seriously infringe body image (BI) [1]. As a definition, BI does not only refer to aesthetical appearance; since a more extensive overview, it is the mental representation of women’s outer appearance, with relevant changes during and after cancer [2]. In addition, it is strongly affected by the overall self-image, which is conceptualized as the match between multiple self-images (i.e., current, ideal, and social self-images) [3]. This way, congruity between various self-images is fundamental to maintain a positive BI, avoiding psychopathological outcomes [4,5]. The loss of one or both breasts, hair loss, and visible scarring are some undesirable appearance-related side effects that damage well-being [6,7]. BI is indeed associated with thoughts and emotions, which affects women’s behaviors strongly even some years after cancer [8,9,10]. It is well-known in the scientific literature that the impact of surgical procedures (i.e., mastectomy and breast-conserving surgery) influences BI perception and satisfaction negatively, in both the short and long-term [11,12]. Notably, studies also highlighted the relevant effect of the type of intervention; for example, radical breast removal, which leaves a permanent and negative mark on the body, can have a strong and negative impact on BI satisfaction in comparison to less radical procedures (e.g., breast-conserving surgery) [11]. In other words, breast cancer interventions play an important role due to their association with many negative bodily changes. As a consequence, women have to struggle with the effects of a less positive self-perception during and after the end of oncological treatment, reducing the overall quality of life [13,14].

Accordingly, current studies show several cognitive and emotional consequences. After cancer, women generally complain about so-called *chemobrain*, a cognitive decline associated with brain intoxication that leads to discomfort in terms of mood disorders, convulsions, and vascular complications [15,16]. Symptoms of *chemobrain* often persist for many years after cancer treatments and women may deal with various difficulties, such as returning to work and other everyday activities. Moreover, *chemobrain* disorders affect cognitive skills and flexibility by complicating abilities in decision-making processes [17]. Decision-making is generally defined as the process of identifying and selecting an option from a set of alternatives based on the preferences of the decision-maker [18]. Specifically, regarding breast cancer survivors, decision-making is strongly relevant because it more fully empowers women, letting them make informed and value-consistent decisions while moving towards favorable health outcomes. On the contrary, impairments in cognitive functions (e.g., memory and attention) are a relevant issue for breast cancer survivors who experience more difficulties in making health decisions [19]. On the other hand, BI dissatisfaction may increase negative emotions, such as body shame, due to the fear of receiving negative evaluations from others [12]. In addition, cancer-related fatigue, which is a state of overwhelming exhaustion over extended periods, is a relevant stress factor in breast cancer survivors [20]. Emotional distress, from feelings of sadness to severe depression, may last for years after cancer therapy completion, impairing quality of life [21,22]. Moreover, breast cancer survivors reported fear of cancer recurrence, which is the most common cognitive and emotional disturbance. Fear of cancer recurrence is indeed associated with worry or concerns about cancer returning/progressing and most frequently endorsed women’s unmet need [23,24,25]. Interoceptive sensations, once ignored, are now perceived with high levels of fear, distress, and worry because they are linked to women’s physical health and threats of oncological illness. Lastly, changes in BI can lead to identity-reframing. Women who constantly perceive their bodies as ill may define themselves as “woman at risk”, rethinking future expectations and personal resources [26]. Of note, breast cancer survivors have to cope with an overall identity restructuration, in which the breast cancer journey is involved.

Starting from the present background, breast cancer survivors need to cope with a new and renovated overall self. Literature conceptualized the self as a system of cognitive and affective self-schemas strongly associated with decisions and life-meanings [27,28]. Thus, a new self-representation related to breast cancer journey, namely *Injured Self*, has to be integrated into the overall self. As a definition, *Injured Self* is an illness-schema characterized by autobiographical memories and emotions related to the oncological journey [29]. The reciprocal interconnection between autobiographical memories and self-schemas is particularly consistent with one’s self-images [30,31]; this way, a positive and integrated *Injured Self* into an overall one is essential to avoid self-fragmentation and promote well-being after cancer [29].

In conclusion, it is paramount to introduce novel and tailored interventions that can promote a positive BI after breast cancer. Particularly, psychological programs should consider patients’ needs and desire, implementing specific and personalized strategies. Starting from the present background, this contribution aims to propose available and tailored psychological interventions that can promote a positive BI in breast cancer survivors.

## 2. Current Psychological Interventions

Managing the individual experiences of illness and, in particular, addressing bodily issues can promote a positive perception of BI after cancer [32]. Current literature demonstrates the effectiveness of mixed-methods interventions based on various approaches (e.g., psychosocial, supportive, cognitive-behavioral/existential, interpersonal, emotionally expressive, and educational programs) to promote well-being and BI [33,34,35]. Notably, a review and meta-analysis study [36] showed a medium, statistically significant effect on BI in breast cancer patients and survivors. Interestingly, authors highlighted that improvements were not modality-specific; BI benefits were strongly connected to tailored interventions on actual women’s needs. This review and meta-analysis indeed demonstrated that various psychological approaches can be implemented, starting from psychological characteristics of each breast cancer survivor. For example, self-compassion and mindfulness-based stress reduction approaches could be helpful for women who show an inability at taking care of their own body and reducing self-judgement [37,38]. On another side, Blanco and colleagues (2021) [33] demonstrated that cognitive-behavioral, interpersonal, educational, and psychosocial approaches decrease psychological distress, promote relaxation, and modify the negative perception of one’s body. At the same time, the current literature showed that both online and in-person interventions can be implemented. To date, research on performing web-based interventions has grown and virtual psychological interventions are even more applied, especially after the COVID-19 emergency [39]. Mifsud and colleagues (2021) [38] demonstrated the efficacy and potential clinical use of a web-based self-compassion-focused writing activity to reduce BI distress in breast cancer survivors. Similarly, the effectiveness of one-on-one psychotherapy, couple intervention, and structured groups to improve BI was also confirmed by literature [38]. Lastly, current literature also promotes psychological interventions to prevent poor BI outcomes after cancer. A review by Fiser and colleagues (2021) [40] synthesized available psychological interventions during preoperative planning, local and systemic treatment, and survivorship to prevent negative BI without compromising oncologic success. To sum up, choosing the appropriate intervention for each survivor and their families by understanding the unique circumstances involved is fundamental [39].

### Cognitive-Behavioral Therapy

The existing evidence-based research on BI prevention and intervention approaches has evolved over time. Cognitive-behavioral therapy (CBT) is one of the most empirically supported interventions to address BI concerns [8]. In general, CBT aims to modify irrational and dysfunctional thoughts, emotions, and behaviors through various techniques (e.g., cognitive restructuring, self-monitoring, desensitization, psychoeducation, and exposure and response prevention) [41]. Concerning BI issues, Cash and colleagues (1997) [42] were among the first researchers to develop and evaluate CBT programs. Authors proposed a psychoeducational intervention focused on causes, prevalence, and effects of BI dissatisfaction: relaxation training, desensitization (e.g., mirror exposure), self-monitoring (i.e., antecedent events, beliefs, consequences), identification and correction of cognitive BI issues, and relapse-prevention strategies. The program effectively decreased body dissatisfaction, anxiety related to aesthetical appearance, and BI-related avoidance, with improvements in the long run. Similarly, Ahmadi and colleagues (2017) [43] demonstrated the effectiveness of a CBT focused on 8-step program to promote BI by challenging and improving women’s irrational beliefs about their bodies. Furthermore, CBT is particularly suggested to address helplessness or hopelessness issues by enhancing one’s ability to cope with life stressors due to cancer diagnosis and treatment [44]. CBT can indeed increase the perception of control in cancer patients, improving emotional balance and overall well-being [45,46]. In other words, a CBT, as based on a time-limited and a goal-oriented approach focused on changing patterns of thinking and behaviors, can prevent future BI disturbances [47].

## 3. Tailored Psychological Interventions: Cognitive, Behavioral, and Emotional Benefits on Bi

As aforementioned, impairments in BI can lead to intense consequences on breast cancer survivors’ well-being. This opinion study focuses on three of the main behavioral issues in breast cancer survivors, proposing specific and tailored psychological interventions (see Table 1). Specifically, we address *checking behaviors*, *social relationships*, *and intimate relationships and sexuality.* Firstly, studies showed that checking the body for signs or symptoms of cancer is a daily behavior that can affect emotional well-being in the majority of breast cancer survivors [48]. Second, social relationships are often affected by negative BI, and even more breast cancer survivors express difficulties in staying in contact with others due to shame and perceived other devaluations [49]. Third, sexuality and intimate relationships are strongly infringed by oncological treatments and their related side effects, which infringe physical contacts and intimate connection with the partner. Thus, breast cancer survivors often report issues in their sexuality after cancer, leading to the risk of breaking off intimate relations [50,51]. Since the present background, the present opinion study proposes available psychological intervention to address and manage *checking behaviors*, *social relationships*, *and intimate relationships and sexuality*, as follows:
-*Checking Behaviors* = fear of cancer recurrence could result from negative body perception and the fear of inner sensations, which are checked as a source of danger and fear [52]. High fear of cancer recurrence led women to perform *checking behaviors* in an obsessive way daily. *Checking behaviors* are defined as excessive breast self-examinations looking for nodules in the breast(s) daily, which promotes distress and anxiety [53]. A review by Tauber and colleagues (2019) [25] reported a small but robust effect of psychological interventions on the fear of cancer recurrence. Particularly, managing processes of cognition, such as worry, attentional bias, and rumination, is essential to change the inner emotional experience. Similarly, a cognitive-behavioral therapy implemented by van de Wal and colleagues (2018) [54] increased the patient’s perceived control over fear of cancer recurrence, who was reduced to a non-clinical level. Improvements were still evident in the long run, specifically at the 6- and 12-month follow-up assessments. Interestingly, authors highlighted that addressing cognitive processes is more helpful than changing the contents of fear of cancer recurrence. In other words, changes in patterns of thoughts and dysfunctional biases are essential to promote a better BI.-*Social Relationships* = visible scarring and hair loss, as some of the main side-related effects of oncological treatments increase body shame and the fear of being different from cultural stereotypes [55]. Thus, BI dissatisfaction triggered by the fear of others’ negative evaluation leads to a strong denial of oneself and low emotional regulation [56]. This is in line with the Self Discrepancy Theory by Higgins (1987) [4], who demonstrated that higher discrepancy between the current (i.e., “Who I am?”) and ideal (i.e., “Who I would like to be?”) self-representations leads to higher psychopathological outcomes. On a behavioral level, negative cognitive beliefs about oneself and the own body increase disgust and anxiety for others’ devaluation, leading to avoiding social relationships [56]. Current studies propose a compassion-focused therapy on treating shame proneness and self-criticism [57]. Particularly for people with chronic disease, compassion-focused intervention can promote psychological adjustment gradually thanks to kindness toward the self, lack of self-judgment, acknowledging past trauma, and accepting suffering as part of the human condition [58,59]. After cancer, a self-compassion approach toward the body, so-called *bodily-self compassion*, is essential to recognize negative emotions and accept uncomfortable feelings without self-judgement and high fear of cancer recurrence [32]. Considering the relevance of body shame and unacceptance of changes, promoting kindness towards one’s new body after cancer can be helpful to promote social connection, decreasing others’ devaluation and fear of judgement. In this regard, scientific literature demonstrated that bodily-compassion intervention in a group could be more effective thanks to benefits associated with the sense of group-belonging, which increase self-esteem and a low sense of isolation [60].-*Intimate Relationships and Sexuality*: breast cancer interventions and their related changes on the body can impair the perception of femininity, intimate relationship, and sexual attractiveness [28,61]. Literature demonstrated that a negative BI could lead to a perceived loss of femininity and body integrity [50]. Women tend to be reluctant to look at themselves naked due to dissatisfaction with surgical scars and feeling less sexually attractive [62]. Accordingly, the loss of breast(s) may have multiple meanings and trigger conflicting emotions about the own perception of femininity and sexual attractiveness [51]. At the same time, it is paramount to consider that oncological treatments increase the risk of incurring premature menopause and infertility, infringing on one’s own womanhood [63]. This way, intimate relationship and sexuality issues request the need of involving partners in the psychological intervention, considering their essential role in providing emotional support, managing finance, and making decisions over their spouses’ cancer treatments, with negative impacts on intimacy and sexuality over the oncological journey [64]. However, not all types of social support are helpful for breast cancer survivors, despite the positive intentions from their partners [65,66]. For example, caregivers may tend to avoid speaking about cancer and its related side-effects; however, breast cancer survivors could have the need to communicate about their past and current fears, sharing the perception of being unattractive or less feminine with their partners [66,67]. On a practical level, a couple’s communication skills training can increase abilities to communicate effectively and openly with each other about cancer-related concerns, improving the quality of intimate relationship and psychological adjustment [65]. At the same time, supporting marital adjustment is fundamental to avoiding their excessive distress and anxiety as caregivers [27,64].

## 4. Conclusions

Women with a history of breast cancer often suffer from negative body perception and dissatisfaction, with notable consequences on their quality of life [68]. Interoceptive sensations are associated with negative emotions due to fear of cancer recurrence, in line with a new illness-schema called Injured Self [29]. Thus, cognitive and emotional impairments after breast cancer can decrease the perception of being autonomous, setting goals, and self-esteem [5]. On a behavioral level, breast cancer survivors may tend to depend on others, promoting unsupportive social and intimate relationships. Similarly, body dissatisfaction affects social relationships and sexuality due to the fear of being negatively evaluated by others and being unattractive to the partner [61]. Thus, it is paramount to increase emotional self-regulation and a positive perception of social support in the long run [69]. To date, literature shows several psychological interventions to promote BI in breast cancer survivors [36]. Of note, the body is always involved in oncological treatments and interventions; however, not all psychological interventions are tailored to women’s needs. It is fundamental to assess the oncological journey of women and individual characteristics and needs to personalize psychological intervention [70,71]. Thus, clarifying and proposing psychological interventions to manage BI issues appropriately is needed.

The present opinion study showed BI issues in breast cancer survivors, exploring cognitive and emotional levels; then, tailored psychological interventions are proposed. Firstly, cognitive-behavioral interventions focused on cognitive processes (e.g., memory and attentional biases) may be helpful in decreasing the fear of cancer recurrence, which is one of the main concerns after cancer [25,54]. CBT is indeed proposed to address dysfunctional patterns of thoughts and behaviors by reducing rumination and negative health behaviors (e.g., impairments in treatment adherence). In general, the present intervention may support breast cancer survivors in focusing attention on their inner sensation without fears.

Second, body compassion interventions in groups may improve kindness toward the own body, decreasing self-judgment [23,72]. As a result, a low level of bodily shame and fear of others’ devaluations could improve social relationships [73]. Bodily compassion interventions are proposed as a possible approach to improve emotional awareness and the acceptance of uncomfortable feelings. On one hand, kindness towards the own body can decrease self-judgement, which increases anxiety and distress strongly. At the same time, the present intervention aims to decrease the relevance of others’ evaluation of their body, being more confident and increasing self-esteem. Lastly, couple communication skill trainings could promote emotional expression and regulation, leading to a more intimate relationship with the partner [65,74]. Sexuality could benefit from a better psychological adjustment of breast cancer survivors and caregivers, who are both involved in the oncological journey [75,76]. Specifically, the present contribution highlighted the relevance of promoting couple intervention to address intimate relationship issues. Couple communication skills training is, for example, a possible strategy to implement in order to increase an open dialogue between partners without fears around expressing personal opinions and needs.

### Limitations and Future Research

The limitations of the present contribution could be the exploration of only three domains of BI interest. However, *checking behaviors*, *social relationships*, *and intimate relationships and sexuality* do not cover all breast cancer survivors’ issues. Future research should explore psychological intervention specifically focused on BI perception and dissatisfaction, for example. Moreover, this opinion study is only focused on the breast cancer survivors’ population. Future research should address other types of cancer, focusing on the specificities of any disease and its consequences on BI. Similarly, future studies should better explore BI characteristics, assessing individuals who received various oncological interventions (e.g., removal of one or both breast(s) and/or adjuvant therapies). It is paramount to note that each oncological treatment and intervention can affect the body differently; thus, assessing the oncological journey is essential to tailor and implement personalized interventions.

## Figures and Tables

**Table 1 ijerph-20-02991-t001:** BI characteristics, its related emotions, and tailored psychological interventions.

	*Consequences on BI*	*Emotional* *Effects*	*Psychological Interventions*	*Expected Results*	*Intervention Novelty*
*Checking* *behaviors*	Body is perceived as a source of danger and fear to check in a continuous and obsessive way	Interoception sensations are experienced as threats of illness, promoting distress and anxiety	Cognitive-behavioral therapies can address cognitive abilities firstly, decreasing worry, attentional bias, and rumination as dysfunctional processes	The experience of inners sensations as bodily information, not only threaths of a breast cancer recurrence, to restore a positive BI	Addressing inners sensations by improving a self-focused attention and promoting bodily perceptions after cancer
*Social* *Relationships*	Undesirable side-effects increase body dissatisfaction and the perception of others’ devaluation	Body shame and the fear of being different from cultural stereotypes lead to a strong denial of oneself and low emotional regulation	Bodily-compassion interventions in groups can improve emotional awareness and the acceptance of uncomfortable feelings supported by others	Improvements in kindness toward the body and lack of self-judgments to reduce the fear of others’ devaluations	Being aware of bodily issues as an obstacle for social relationships. Thus, promoting kindness and love for the own body is proposed
*Intimate relationships* *and sexuality*	Women experience their body as unattractive and less feminine	Fear and worry about cancer-related contents, such as sexuality and fertility issues	Couple communication skills trainings can increase open and effective communication with partners	Better intimate relationships and psychological adjustments of both breast cancer survivors and their partners	The perception of a new body after cancer is an essential point to address during a couple intervention

## Data Availability

Not applicable.

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
