# Peer review of "Tailored Psychological Interventions to Manage Body Image: An Opinion Study on Breast Cancer Survivors"

_ijerph, 2023, doi:10.3390/ijerph20042991_

Round 1
Reviewer 1 Report
Introduction
· Page 1, line 37: please clarify if the consequences here refer to the consequences of a breast cancer diagnosis, interventions, or a negative body image.
· Page 1, lines 42-44: This statement is vague. Please elaborate more about the relevance of impairments in decision-making to breast cancer survivors. For example, are there certain aspects or experiences of breast cancer survivors that make impairments in decision-making particularly relevant to this population, more so than to people without breast cancer?
· I would also be interested in reading more about how a negative body image may influence a woman’s self-perception in relation to femininity, especially since authors discussed more about this later in the paragraph about intimate relationships and sexuality.
· It may be helpful to include 1-2 sentences about the objectives of this paper at the end of the introduction.
Current Psychological Interventions
· Please include more details in this section, such as a brief introduction to each of the intervention approaches, more details about the effectiveness of these approaches (e.g., effect sizes), if certain approaches are more effective than others, if some approaches are more effective among certain groups of breast cancer survivors, specific aspects of the body image that these approaches are effective, etc.
Tailored Psychological Interventions
· I wonder why authors choose to focus on these three behavioral domains (i.e., why these three behaviors are more significant than others). Some rationales may be helpful for the audience to understand the context.
· Please provide more information about these interventions and explanations regarding in what ways they are tailored/personalized. Current descriptions are very generalized. For example, CBT is a widely used treatment modality for various psychopathologies. How do authors plan to tailor CBT to meet the unique needs of breast cancer survivors, and more specifically, address the body-checking behaviors?
Conclusion
· Authors may consider editing the conclusion and making it more succinct.
Author Response
Dear Reviewer,
Thank you for your time and precious suggestions on our Manuscript. We hope that the revisions improved the quality of the manuscript. Please note that any modifications to the manuscript have been highlighted in green. Please, see detailed responses below and in the new version of the manuscript. In addition, we adjusted the Manuscript as follows:
INTRODUCTION: firstly, we clarified that breast cancer treatments can have a strong impact on BI in breast cancer survivors, as requested. In particular, impairments in self-perception have been introduced as a new relevant construct to consider. Second, decision-making has been defined; in additions, details were introduced in reference to breast cancer survivors specifically. As presented in the contribution, breast cancer survivors’ decision-making in essential to make healthy decisions, which is relevant in the oncological field particularly. In line with this, we highlighted the association between “chemobrain” and cognitive effects in breast cancer survivors, such as reducing memory and attention, which can affect decision-making. Third, in the paragraph related to “intimate relationship and sexuality”, we specified the strong association between a negative BI and reductions in women’ femininity and sexuality. Current literature showed that bodily self-dissatisfaction after oncological treatments could be the results of reluctance to look at the own self naked due to scars and other side-effect of oncological treatments (e.g., skin discoloration and loss of breast(s)). In this regard, scientific studies have been added (see: Koçan et al., 2016; Fobair et al., 2006; Paterson et al., 2016; Sankar et al., 2022). Lastly, few lines have been introduced to explain study’s objective at the end of the introduction paragraph. We literally stated: “Starting from the present background, this contribution aims to propose available and tailored psychological intervention that can promote a positive BI in breast cancer survivors”.
CURRENT PSYCHOLOGICAL INTERVENTION: thank you for your helpful suggestion to promote this paragraph. As you can see in the Manuscript, we added information about the effectiveness of specific psychological interventions and modalities of approach. In general, it is paramount to consider that literature demonstrated the effectiveness of various psychological intervention (Sebri et al., 2021). Individuals’ characteristics and needs have to be considered in order to propose the best psychological intervention’ option. Nevertheless, studies demonstrated that some type of psychological could be strongly effective to address a specific breast cancer issue. Thus, we focused on cognitive behavioral therapy, body-compassion intervention, and couple communication skills training. Particularly, literature evidenced the essential role of cognitive behavioral therapy as one of the most empirically supported approach to address BI issue; in this regard, a new sub-healing about it has been introduced (see 2.1 Cognitive-behavioral therapy paragraph).
TAYLORED PSYCHOLOGICAL INTERVENTION: thank you for your request of promoting this manuscript’ paragraph. Firstly, we clarified why checking behaviors, social relationships, and intimate relationships and sexuality have to be addressed in a breast cancer survivors’ population, as requested. Starting from the current scientific literature, these domains are three of the main behavioral issues that breast cancer survivors have to deal with. Specifically, checking the body due to scars and/or negative interoceptive sensations is a daily behavior that can affect emotional well-being strongly (Soriano et al., 2019). Second, breast cancer survivors often express difficulties in staying in relationship with others due to body shame and fear of devaluations (Ashaq et al., 2019). This is in line with intimate and sexuality issues, which result from bodily changes and their perception of body unacceptance by breast cancer survivors (please, see at the beginning of this paragraph). Of note, checking behaviors, social relationships, and intimate relationships and sexuality are three of the main behavioral issues to cope for breast cancer survivors; however, other important relevant challenges have to be considered. We add this consideration in the limitation paragraph. Second, we explained in depth the results of the three available psychological intervention (cognitive behavioral therapy, body-compassion intervention, and couple communication skills training), providing specific information regarding each of them. As previously stated, see also the sub-healing titled cognitive-behavioral interventions for more details, please.
CONCLUSION: conclusions have been adjusted also considering the third reviewer suggestion. Please, feel free to highlight unnecessary and redundant lines of the text, if present.
References
- Fobair, P., Stewart, S. L., Chang, S., D'Onofrio, C., Banks, P. J., & Bloom, J. R. (2006). Body image and sexual problems in young women with breast cancer. Psycho‐Oncology: Journal of the Psychological, Social and Behavioral Dimensions of Cancer, 15(7), 579-594. https://doi.org/10.1002/pon.991
-Garcia, S. N., Coelho, R. D. C. F. P., dos Santos, P. N. D., Maftum, M. A., de Fátima Mantovani, M., & Kalike, L. P. (2017). < b> Changes in social function and body image in women diagnosed with breast cancer undergoing chemotherapy. Acta Scientiarum. Health Sciences, 39(1), 57-64. https://doi.org/10.4025/actascihealthsci.v39i1.31833
-Kieszkowska-Grudny, A., Rucinska, M., Ciesak, R., & Wisniewska, M. (2017). Body image, self-esteem and quality of life during oncology treatment in breast cancer, cervical cancer, and prostate cancer in comparison to the healthy population.
- Koçan, S., & Gürsoy, A. (2016). Body image of women with breast cancer after mas-tectomy: a qualitative research. The journal of breast health, 12(4), 145. 10.5152/tjbh.2016.2913
-Kołodziejczyk, A., & Pawłowski, T. (2019). Negative body image in breast cancer patients. Advances in clinical and experimental medicine: official organ Wroclaw Medical University, 28(8), 1137-1142. 10.17219/acem/103626
- Padamsee, T. J., Wills, C. E., Yee, L. D., & Paskett, E. D. (2017). Decision making for breast cancer prevention among women at elevated risk. Breast Cancer Research, 19(1), 1-12. 10.1186/s13058-017-0826-5
Rezaei, J. (2016). Best-worst multi-criteria decision-making method: Some properties and a linear model. Omega, 64, 126-130. https://doi.org/10.1016/j.omega.2015.12.001
- Paterson, C., Lengacher, C. A., Donovan, K. A., Kip, K. E., & Tofthagen, C. S. (2016). Body image in younger breast cancer survivors: a systematic review. Cancer nursing, 39(1), E39. 10.1097/NCC.0000000000000251
- Sankar, U. V., Warrier, N. E., TS, R., & Wilson, R. (2022). Understanding women’s perspectives on breast cancer is essential for early breast cancer diagnosis and to avoid diagnosis delay: A mixed method study. http:// 10.1200/JCO.2022.40.16_suppl.e240
- Sebri, V., Durosini, I., Triberti, S., & Pravettoni, G. (2021). The efficacy of psychological intervention on body image in breast cancer patients and survivors: a systematic-review and meta-analysis. Frontiers in Psychology, 12, 611954. https://doi.org/10.3389/fpsyg.2021.611954
- Van Dyk, K., & Ganz, P. A. (2021). Cancer-related cognitive impairment in patients with a history of breast cancer. JAMA, 326(17), 1736-1737. Htttp:// 10.1001/jama.2021.13309
Reviewer 2 Report
The contribution of psycho-oncology to the well-being of cancer patients leaves no doubt. In this short article the authors broaden the concept of body image (BI) to incorporate cognitive and affective factors and consider BI from a physical, cognitive, emotional, and social perspective. Aimed specifically at women with breast cancer, they propose different psychotherapeutic approaches to help reconstruct this holistic body image from the sequelae of the disease.
As an opinion article, it fits perfectly within the theme of the special issue New Theoretical Frameworks and Psychological Interventions in cancer and is an interesting contribution to the mind-body approach in health sciences.
Table 1 is very useful to understand the article.
Question:
It seems to me that body image, as discussed in the article is equivalent to self-image, is that the authors intention? Please specify

Author Response
Dear Reviewer,
Thank you for your work on our manuscript and for the positive comments. Starting from your suggestion, we improved this contribution as depicted below. Kindly notice that any modification to the manuscript has been highlighted in green to aid consultation of review.
As requested, we clarified differences between Body Image and Self-Image at the beginning of the introduction section). As reported by the current literature, Body Image refers to the cognitive and mental representation of one’s external appearance, with a strong and intensive association with emotions and behaviors (Hale et al., 2015). Particularly, in a study by Hale and colleagues (2015), participants evidenced the impact of negative evaluations on their Body Image, also in terms of outward physical appearance. Visible skin changes, scars, and weight gain can cause strong distress as much as they felt their outward appearance negatively changed. At the same time, Body Image can be strongly affected by how you believe other people view (public self-consciousness) (Hale et al., 2015). On the other side, Self-Image is conceptualized as the match between multiple self-images about the own Self. For instance, individuals experience differences or similarities between who they are (actual Self-image), who they would like to be (ideal Self-Image), and others ‘evaluations (social Self-Image) (O’Cass et al., 2015). Thus, discrepancies between different self-images reduce a positive emotional well-being (Higgins, 1987). Accordingly, literature highlighted the relevance of Self-Image congruity in order to avoid self-fragmentations and, as a consequence, emotional and behavioral issues (Li et al., 2020).
References:
- Canevello, A., & Crocker, J. (2015). How self‐image and compassionate goals shape intrapsychic experiences. Social and Personality Psychology Compass, 9(11), 620-629.
- Hale, E. D., Radvanski, D. C., & Hassett, A. L. (2015). The man-in-the-moon face: a qualitative study of body image, self-image and medication use in systemic lupus erythematosus. Rheumatology, 54(7), 1220-1225.
- Higgins, E. T. (1987). Self-discrepancy: a theory relating self and affect. Psychological review, 94(3), 319.
- Li, S., Wei, M., Qu, H., & Qiu, S. (2020). How does self-image congruity affect tourists’ environmentally responsible behavior?. Journal of Sustainable Tourism, 28(12), 2156-2174.
- O’Cass, A., & Muller, T. (2015). A study of Australian materialistic values, product involvement and the self-image/product-image congruency relationships for fashion clothing. In Global Perspectives in Marketing for the 21st Century: Proceedings of the 1999 World Marketing Congress (pp. 400-402). Springer International Publishing.
Reviewer 3 Report
The proposed theme is a topical one. The review of some of the cognitive-behavioural aspects involved is relevant. Perhaps more relevant would have been to analyze in a paragraph the most effective forms of CBT.
It is not clear how exactly you selected the studies you write about.
A comparison of CBT types would have been beneficial. You have not analysed helplessness and hopelessness. Perhaps you will manage to raise the level of the article in this way. Just a suggestion.
Table 1 is a summary. That's good. However, what is new compared to the established literature in the field. Introduce explanations that make the most of the table.
The opinions of the authors seem to become clear only in the part devoted to conclusions.
Conclusions should be developed.
These are just suggestions that I think would increase the value of your work.
Author Response
Dear Reviewer,
thank you for your evaluation and your efforts to promote our Manuscript. We adjusted the Manuscript following your suggestions. Please note that any modifications to the manuscript have been highlighted in green. Thank you again for the time you dedicated to our contribution.
Firstly, we added a sub-healing regarding the most effective forms of cognitive behavioral interventions (see in the current psychological interventions section). Specifically, in this paragraph we added more details about cognitive behavioral intervention’ definitions and its possible implications on Body Image issues (Lewis-Smith et al., 2019; McMain et al., 2015). We clarified that a time-limited and goal-oriented intervention can effectively address and change dysfunctional thoughts and behaviors related to BI (Cassone et al., 2016; Monteiro-Guerra et al., 2020). Additionally, effectiveness of some research studies has been introduced (see Ahmadi et al., 2017; Cash et al., 1987), also in reference to hopelessness and helplessness, as suggested. Specifically, in the current psychological interventions section, we highlighted the relevant role of CBI in reducing hopelessness and helplessness in cancer patients, who suffer with life stressors because of cancer diagnosis and treatments (Gautam et al., 2020; peoples et al., 2019; Vizin et al., 2019). Second, concerning the presented studies, we selected current research contributions that showed efficacy and helpful techniques to cope BI issues (checking behaviors, social relationships, and intimate relationships and sexuality particularly). As you can see in both “current psychological interventions” and “tailored psychological interventions: cognitive, behavioral, and emotional benefits on BI” paragraph, details about psychological intervention characteristics and effectiveness have been presented. Third, we included a new column in Table 1 to better explain the relevance of the proposed psychological interventions. Particularly, we specified the novelty features related to the proposed programs of intervention focused on the promotion of BI. Fourth, we clarified our opinions also in the introduction section, as requested. In particular, we specified the need of personalized psychological intervention to promote a positive BI after cancer. At the same time, we highlighted that breast cancer survivors have to deal with an overall identity changes, which is a fundamental aspect to address in future research. Lastly, conclusions have been increased by adding details about each specific topic of interest that are involved in the promotion of BI. Accordingly, we made a new sub-healing titled “limitations and future research” to improve the article’ relevance and structure.
References
- Ahmadi, Z., Abbaspoor, Z., Behroozy, N., & Malehi, A. S. (2017). The effects of cognitive behavioral therapy on body image in infertile women. Iranian Red Crescent Medical Journal, 19(10). http:/7doi.org7 10.5812/ircmj.14903
- Cash, T. F. (1997). The body image workbook: An 8-step program for learning to like your looks. New Harbinger Publications, Inc.
- Cassone, S., Lewis, V., & Crisp, D. A. (2016). Enhancing positive body image: An evaluation of a cognitive behavioral therapy intervention and an exploration of the role of body shame. Eating disorders, 24(5), 469-474. https://doi.org/10.1080/10640266.2016.1198202
- Gautam, M., Tripathi, A., Deshmukh, D., & Gaur, M. (2020). Cognitive behavioral therapy for depression. Indian journal of psychiatry, 62(Suppl 2), S223.
- Lewis-Smith, H., Diedrichs, P. C., & Halliwell, E. (2019). Cognitive-behavioral roots of body image therapy and prevention. Body Image, 31, 309-320. https://doi.org/10.1016/j.bodyim.2019.08.009
- McMain, S., Newman, M. G., Segal, Z. V., & DeRubeis, R. J. (2015). Cognitive behavioral therapy: Current status and future research directions. Psychotherapy Research, 25(3), 321-329. https://doi.org/10.1080/10503307.2014.1002440
- Peoples, A. R., Garland, S. N., Pigeon, W. R., Perlis, M. L., Wolf, J. R., Heffner, K. L., ... & Roscoe, J. A. (2019). Cognitive behavioral therapy for insomnia reduces depression in cancer survivors. Journal of Clinical Sleep Medicine, 15(1), 129-137.
- Vizin, G., & Farkas, K. (2019). Possibilities of cognitive behavioral therapy in the oncological care. Magyar Onkologia, 64(1), 62-69.
Round 2
Reviewer 1 Report
Thank you for your response to my comments! I enjoyed reading the revised manuscript. I particularly appreciated the additional rationales provided in relation to the unique experiences of breast cancer survivors. This has greatly improved the novelty and significance of this study.